# Online model selection by learning how compositional kernels evolve

**Eura Shin**  *eurashin@g.harvard.edu*
*Department of Computer Science*
*Harvard University*

**Predrag Klasnja**  *klasnja@umich.edu*
*School of Information*
*University of Michigan*

**Susan A. Murphy**  *samurphy@g.harvard.edu*
*Department of Computer Science*
*Harvard University*

**Finale Doshi-Velez**  *finale@seas.harvard.edu*
*Department of Computer Science*
*Harvard University*

**Reviewed on OpenReview:** *https://openreview.net/forum?id=23WZFQBUh5*

## Abstract

Motivated by the need for efficient, personalized learning in mobile health, we investigate the problem of online compositional kernel selection for multi-task Gaussian Process regression. Existing composition selection methods do not satisfy our strict criteria in health; selection must occur quickly, and the selected kernels must maintain the appropriate level of complexity, sparsity, and stability as data arrives online. We introduce the Kernel Evolution Model (KEM), a generative process on how to evolve kernel compositions in a way that manages the bias-variance trade-off as we observe more data about a user. Using pilot data, we learn a set of *kernel evolutions* that can be used to quickly select kernels for new test users. KEM reliably selects high-performing kernels for a range of synthetic and real data sets, including two health data sets.

## 1 Introduction

Online, multi-task learning is common in many machine learning applications, ranging from recommender systems (Luo et al., 2020) to online education (Sayed et al., 2020). In these settings, pools of available data grow *for each user* as they continue to interact with the application platform; a single *task* consists of maintaining the best model for an individual at any given time. We are specifically motivated to model online, multi-task problems in *mobile health (mHealth)*. For example, in mobile fitness, we want to model a user's daily activity levels given other mobility data to plan the delivery of personalized messages. Each user's data is a time series in which model outputs are needed, online, in order to personalize health interventions.

Mobile health applications have a number of domain-specific assumptions and constraints. For our applications, we can assume the availability of pilot training data (on the order of tens to hundreds of users Lee et al. (2016); Cafazzo et al. (2012); Choi et al. (2016)) that may be used to prepare a model for deployment in a high-stakes clinical trial. We can assume a generous pre-deployment period to work with the pilot data (on the order of months to years) and to commit to a set of modeling decisions (Trella et al., 2022). To maintain the integrity of the study during deployment, our models are held to **strict criteria**; we must be able to *update* the model for each user as data arrives online, and the updated model must be *appropriately*

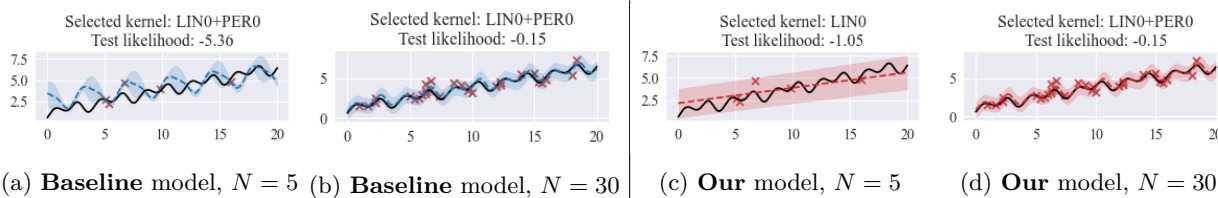

(a) **Baseline** model, $N = 5$ (b) **Baseline** model, $N = 30$ (c) **Our** model, $N = 5$ (d) **Our** model, $N = 30$

Figure 1: **Our selection model (in red, fig. 1c and fig. 1d) infers the kernel composition for a new user and adjusts its complexity given more data, $N$.** We demonstrate the quality of the kernels selected by our method (red) versus the baseline method (blue) by plotting the GP predictions given varying numbers of observations, shown in red "X"s. We want the selection method to choose kernels that result in good predictions when we have observed little data (fig. 1a, fig. 1c) and sufficient data (fig. 1b, fig. 1d). The GP mean predictions are dotted lines and the uncertainties are shaded; a good prediction is as close as possible to the true function (the solid, black line) and has high test likelihood.

*complex* and *interpretable.* By appropriate complexity, we mean that the model should not overfit when there is initially a small amount of data per user, or underfit as the available data grows; in mHealth, a poorly fit model risks sending inappropriate interventions that lead to user dropout. By interpretable, we mean that the model must predict *uncertainties*, include only a *sparse* number of relevant features, identify the *structural relationship* of the feature to the predictions, and maintain *stability* in the features and relationships between updates. Such characteristics allow for verification of the model and scientific discovery of meaningful relationships from the data.

Since Gaussian Processes naturally handle scarce, noisy data with the capacity to model complex trends and their uncertainties, they are a good candidate to meet such interpretability standards in health (Tomkins et al., 2020; Cheng et al., 2020; Ghassemi et al., 2015). As a result, we focus on GP regression for modeling online health data. However, a poor choice of kernel is detrimental to GP performance (Oyetunde & Liem, 2022; Stephenson et al., 2021), and current kernel selection approaches do not meet the strict needs– adaptive complexity and interpretability– of our multi-task, online health settings.

For single-task learning, compositional kernel search (CKS) methods use arithmetic combinations of simple kernels to encode structures in data (Duvenaud et al., 2013). Structural kernels selected by CKS are sparse and encode relationships that are intuitive to understand (Lloyd et al., 2014; Schulz et al., 2017). However, restarting the selection procedure from scratch risks losing the stability of the structure: the kernel included in one time step may differ in the next. Furthermore, these methods do not scale to the update cadence that is required for multi-task settings—we need to repeat this expensive search every time a new user joins the app or the model for an existing user is updated. With multiple users (tasks), we can share information about the best kernel across users. CKS can be made scalable through multi-task learning, in which a small set of representative kernels is selected on pilot users and then used for modeling new users in general deployment. However, existing multi-task approaches consider the *final* data sets of pilot users when identifying representative kernels (Titsias & Lázaro-Gredilla, 2011). For online selection, directly transferring a complex kernel composition, learned on a pilot user with many data points, to a new user with few points leads to overfitting (see fig. 1a).

Our method for choosing representative kernels explicitly addresses the challenges of online, multi-task model selection in mHealth. We increase the complexity of the kernel only when demanded by data. To do so, we learn the best *sequences* of kernels that will balance the bias-variance trade-off for users with growing amounts of data (see fig. 1c and fig. 1d). These sequences, which we call *kernel evolutions*, encode the relationship between former and current kernels, which can be transferred from pilot to test users. By design, our kernel evolutions encode stability and sparsity, since they demonstrate how user compositions change from one time step to the next.

We illustrate the efficacy of our approach on synthetic examples, UCI datasets, and real health data selected to match the characteristics of data we would find in mHealth. When data arrive online, our method selects kernels that are (1) **sparse**, including only features that are relevant to the prediction target; (2)

**stable**, includes the same kernel components (i.e. features) consistently across time steps; (3) of **adaptive complexity**, selecting kernels that manage the bias-variance trade-off.

## 2 Related Work

**Batch Compositional Kernel Selection.** Our compositional kernel selection strategy is specifically designed for online settings. That said, one way to use *batch* selection methods for online data is to re-run the selection algorithm on the user's cumulative data set each time new data arrives. The Automatic Statistician literature describes batch methods that search over a grammar of compositions (Steinruecken et al., 2019). Duvenaud et al. (2013), and Lloyd et al. (2014) introduced Compositional Kernel Search (CKS), a greedy search over the sums and products of simpler base kernels. Recent work has made CKS more scalable by approximating the model evidence (Kim & Teh, 2018) or representing the search as a neural network (Sun et al., 2018). The greedy method of Duvenaud et al. (2013) is most similar to our own, in that it prioritizes building the strongest kernels from the data into the composition. However, the space of compositions these batch methods must search over is already extremely large (Gardner et al., 2017), and infeasible to repeat from scratch each time new data arrives.

**Online Composition Selection.** Probabilistic composition selection methods (Malkomes et al., 2016; Gardner et al., 2017; Titsias & Lázaro-Gredilla, 2011; Tong et al., 2021; Zhang et al., 2019) are better suited for online data since one can use the previous posterior over kernels as the updated prior for training on new data. For example, Tong et al. (2021) uses shrinkage priors to select a sparse subset of kernel components.

A recent subset of CKS methods attempt kernel selection specifically for non-stationary functions– online data whose generating kernel changes over time. Adaptations of Duvenaud et al. (2013) use a "sliding window" on the data and a change-point detection algorithm for initializing CKS (Hüwel et al., 2021; 2022). These methods are more scalable in time but not in users, as they still require many iterations of re-sampling or re-approximating the updated posterior per user. In contrast, our method scales to new users by directly transferring the compositions learned from pilot users.

Finally, there are areas of online (multi) kernel selection that are distinct from online *compositional* kernel selection. Some online methods are designed specifically for Support Vector Machines (Zhang & Liao, 2018; Yang et al., 2012; Orabona et al., 2010; Hoi et al., 2013), which lack predictive uncertainties and bypass dealing with the cubic time complexity of calculating a GP marginal likelihood. Lu et al. (2020) use random features to approximate an ensemble of stationary GPs but does not describe how to combine different base kernels (e.g. linear and periodic), sparsely select over components, or search over hyperparameters. Similarly, Zhang et al. (2019) learn a mixture of (infinite) GPs that evolves over time, but the GPs that they consider have the same kernel composition and differ by the hyperparameters; they do not describe how to jointly search over compositions and their hyperparameters. Finally, Levi & Ullman (2009) poses a model-agnostic algorithm that incrementally increases the model's capacity but does not address how to search over a large composition space. In summary, these works are missing at least one key element of compositional kernels for GPs, which are desirable for their interpretability in our setting.

**Multi-task Selection.** One way to make CKS amenable to multi-task settings is by leveraging kernel similarities across tasks. Methods from Tong & Choi (2019) and Titsias & Lázaro-Gredilla (2011) learn a set of kernel compositions on multiple users that can be transferred to new users. However, these approaches risk overfitting, as they do not restrict the complexity of the composition that is being transferred.

KEM's approach of transferring kernel evolutions from a set of pilot users to a set of test users is related to the meta-learning paradigm. Classical MAML (Finn et al., 2017) and its variants (Yoon et al., 2018; Finn et al., 2018) require differentiable loss functions– relaxing our *discrete* search over compositions to a *soft* search would reduce sparsity. Similarly, non-compositional approaches from Garnelo et al. (2018); Rothfuss et al. (2021a;b) sacrifice the sparsity and stability that our additive compositions provide.

## 3 Background

**Compositional kernels.** We assume that the reader is familiar with GP regression (Rasmussen & Williams, 2006). Compositional kernels express complex functions through sums and products of simpler kernels Duvenaud et al. (2013). The sum of two functions with independent priors, $f_1 \sim GP(0, k_1)$ and $f_2 \sim GP(0, k_2)$, corresponds to the same operations on the kernels: $f_1 + f_2 \sim GP(0, k_1 + k_2)$. Furthermore, the product of two kernels that are defined on different dimensions of the data allows us to account for interactions between the dimensions. Compositional kernel selection uses these rules to form an expressive grammar over functions. For example, functions that can be recovered with a compositional kernel include generalized additive models ($\sum_{d=1}^{D} \text{SE}_d$) and automatic relevance determination ($\prod_{d=1}^{D} \text{SE}_d$), where $\text{SE}_d$ is a squared exponential kernel on the $d$-th dimension of the data.

**Dirichlet process mixture models.** A Dirichlet Process (DP) mixture model generalizes the Dirichlet distribution from being a conjugate prior over a *fixed* number of clusters to an *infinite* number of clusters (Li et al., 2019). The generative process for data under the DP is: $H \sim DP(\alpha, G), \quad \theta_n | H \sim H, \quad X_n | \theta_n \sim p(X_n; \theta_n)$, where $X_n$ is the observation, $\theta_n$ are the parameters of the distribution that generated $X_n$, and $H$ is a distribution over the parameters ($G$ is the base distribution). The Dirichlet Process assumes $H$ is *discrete*. Consequently, all $X_i, X_j$ with the same $\theta_i = \theta_j$ can be thought of as belonging to the same cluster. Let $\{\theta_c\}_{c=1}^{C}$ represent the $C$ clusters that exist among the observations and $Z_n \in \{1, \ldots, C\}$ be the assignment of an observation to one of these clusters.

One can track cluster assignments under this DP using a Chinese Restaurant Process (CRP). Given the current cluster assignments of all other observations, the $n$-th observation is assigned to cluster $c$ with the following probability:

$$p(Z_n = c | Z_1, \ldots, Z_{n-1}) = \begin{cases} \frac{\alpha}{N + \alpha - 1}, & \text{if } c = C + 1 \\ \frac{N_c}{N + \alpha - 1}, & \text{otherwise} \end{cases}, \tag{1}$$

where $N_c$ is the number of observations currently assigned to the $c$-th cluster of $N$ total observations. Under this model, the probability of starting a new cluster is determined by $\alpha$. The probability of assignment to an existing cluster is proportional to the number of observations already in that cluster. For more information on DPs, see Teh (2010).

## 4 Problem Setting

During deployment, user data arrives online – in batches over multiple time steps – and we want to efficiently select the "best" kernel for every user at every time step. Below, we define our kernel notation and what it means to select the "best" kernel for a user at a given point in time.

**Kernel notation.** We denote kernel compositions as $K$, kernel hyperparameters (e.g., lengthscale, period) as $\theta$, and kernels (combination of a composition **and** hyperparameters) as $K_\theta$. Bold notation represents a set; for example, $\boldsymbol{K}$ is a set of kernel compositions. We attribute any of these entities to a user $u$ and/or time $t$ through subscripts. For example, $K_{\theta_{u,t}}$ is a kernel and $\boldsymbol{K}_{u,t}$ is a set of kernel compositions for the $u$-th user at time $t$.

**Kernel compositions.** As in Tong et al. (2021), we define a kernel composition as a weighted sum of "candidate kernels":

$$K(x, x') = \sum_{i=1}^{I} w_i K_i(x, x'), \tag{2}$$

where $k_i$ is one of $I$ candidate kernels and $w_i > 0$ is the weight.

**Formalizing online model selection.** We begin by defining what it means for the user's data to arrive online. Let $\mathcal{D}_{u,t} = \{\mathbf{X}_{u,t}, \mathbf{y}_{u,t}\}$ represent the *cumulative* data set of all observations for user $u$ up to time $t$. We assume that the user's data is generated by a fixed, latent function $f_u(\mathbf{x}) : \mathbb{R}^D \to \mathbb{R}$ for $D$ dimensional inputs $\mathbf{x}$. We assume $f_u$ was sampled from a GP with an unknown kernel. The observed targets are corrupted by a user-specific level of noise, such that $y = f_u(\mathbf{x}) + \epsilon_u$ for $\epsilon_u \sim \mathcal{N}(0, \sigma_u^2)$.

Now we are prepared to define what it means to select the best kernel for a user's online data. At time $t$, we must select a kernel composition $K_{u,t}$ and hyperparameters $\theta_{u,t}$ so that $\hat{f}_{u,t} \approx f_u$ where $\hat{f}_{u,t} \sim GP(\cdot, K_{\theta_{u,t}})$. Using our observations from the user up to this point $\mathcal{D}_{u,t}$, the kernel $K_{u,t}$ can be selected by maximizing the marginal likelihood, or, as we will later do, by minimizing the BIC for a restricted set of compositions.

**Stationarity assumptions.** We make two stationarity assumptions on (1) the distributions between pilot and deployment users; and (2) the data distribution within a single user. First, we assume there is no distribution shift between pilot users and deployment users. In health, the pilot trial is designed to target the same population as in deployment. For example, an earlier and follow-up version of the HeartSteps trial targeted adults newly diagnosed with stage 1 hypertension in the Seattle area (Ghosh et al., 2023). Second, we assume that the data-generating function for a single user, $f_u$, does not change over time. The evolution of the "best" composition between time steps **does not** indicate non-stationarity, but rather, our *adjustments to the bias–variance trade-off* in selecting the best-fitting model for the data we currently have about a user. In health settings, stationarity is often assumed because the noisiness and sparsity of the data make it difficult to directly model non-stationary effects online (Trella et al., 2022).

## 5 Kernel Evolution Model (KEM)

**Problem statement.** During deployment, we must select kernel compositions for new users in a manner that is of appropriate complexity to the currently available data, and interpretable– sparse and stable– in the components that are included.

**Our solution.** We propose the Kernel Evolution Model (KEM), a Bayesian non-parametric model that describes how a user's kernel evolves as a function of observing additional data. Our solution happens in two stages, pilot training and deployment testing, which we describe in section 5.1. We use KEM to find a set of *kernel evolutions* offline (pilot training, section 5.3), which enables us to efficiently select kernel compositions online (deployment testing, section 5.4) in a sparse and stable manner.

### 5.1 Pilot training vs. deployment testing.

*Example timeline.* KEM learns how to select kernels during a long pilot training phase in order to efficiently select kernels for new users during deployment. In pilot training, we have the time and resources to perform any necessary search operations over compositions. For example, there was over one year between the pilot trial of HeartSteps V1, a micro-randomized trial for mobile health activity notifications (Klasnja et al., 2019), and HeartSteps V2 (Liao et al., 2020). In contrast, during deployment testing, new users and their data arrive online, and we must efficiently select kernels for these users with initially limited data. For example, the kernel selection must occur daily (Liao et al., 2020) or weekly (Trella et al., 2022), in a stable manner, for all users. We do not re-learn the set of kernel evolutions between the arrival of test users, so that kernel selection remains stable during deployment.

*Pilot training.* Before the main study, we use data from a pilot study to identify sequences of kernel compositions– which we call kernel evolutions– that manage the bias–variance trade-off as the amount of data increases, per user, for different users. To do so, we process the training users' data incrementally, as though it were arriving "online." For example, for some users, a linear kernel might best model the initial data, but at a later time, a linear + periodic may be best; for others, a squared-exponential kernel may always be preferred. In section 5.2 and section 5.3 we detail the pilot training process that identifies a set of kernel evolutions– with emphasis on sparsity and stability– among the training users.

*Deployment testing.* During deployment, we use the previously learned evolutions to omit compositions that are unlikely to fit, too complicated, not sparse, and not stable. In the above example, if a test user's data is currently modeled by a linear kernel, given new data we might consider either a linear or a linear+periodic kernel, but not try a squared-exponential. (Note: while the selection model helps us efficiently select a *kernel composition* for a new user, the *hyperparameters* of the composition must still be optimized to that test user). In section 5.4 we detail how evolutions are used to select kernels for a new test user.

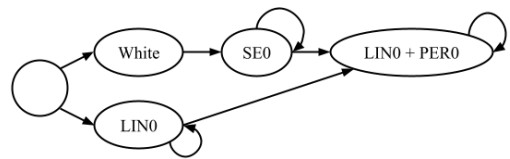

Figure 2: **Example visualization of a selection model learned by KEM on synthetic data.** This directed graph is one way to represent the $K_{\text{parent}} \to K_{\text{child}}$ relationships that are learned in pilot training. Each node in the graph is a DP. The arrows leaving from each node represent the "clusters" found at this DP. It is possible for two separate DP to identify clusters with the same composition; this is why SE0 and LIN0 both have arrows leading to LIN0 + PER0.

### 5.2 KEM: A Generative Process for Online Data

**Modeling evolutions of kernel compositions with a Dirichlet Process.** For a user $u$, we model the evolution of a single composition from $K_{u,t-1}$ to $K_{u,t}$ with a DP mixture model. Generically, we refer to an evolution as a transition of the form $K_{\text{parent}} \to K_{\text{child}}$. Since the distribution over $K_{\text{child}}$ is only conditioned on its immediate predecessor $K_{\text{parent}}$, our process is Markovian– not hierarchical– in how it represents evolutions over time.

*Why a mixture model?* We will learn the set of evolutions $K_{\text{parent}} \to K_{\text{child}}$ by observing the evolutions that occurred among all our pilot users. It is possible that different pilot users, who started with the same $K_{\text{parent}}$, required a different $K_{\text{child}}$ composition. Each cluster in our mixture model corresponds to a unique composition for $K_{\text{child}}$, and each cluster is populated with instances of the given evolution occurring among the pilot users. *Why a DP?* Using a DP allows us to avoid having to pre-specify the number of clusters– the number of unique compositions ($K_{\text{child}}$) per $K_{\text{parent}}$ that will exist.

Concretely, for every kernel composition $K_{\text{parent}}$, there is a DP over the composition of $K_{\text{child}}$:

$$H_{K_{\text{parent}}}(K_{\text{child}}) \sim DP(\alpha, G_{\text{parent}}|K_{\text{parent}}), \tag{3}$$

where $H_{K_{\text{parent}}}$ defines a distribution over the next composition leaving from $K_{\text{parent}}$. $G_{\text{parent}}$ is the base distribution (whose support is over kernel compositions) and $\alpha$ is the concentration parameter of the DP.

We will now discuss how our choice of $G_{\text{parent}}$ encourages sparsity and stability in the evolution from $K_{\text{parent}}$ to $K_{\text{child}}$. Recall from eq. (2) that a kernel composition is a weighted sum over candidate kernels, where $w_i$ is the weight on the $i$-th candidate kernel. $G_{\text{parent}}$ uses a spike-and-slab distribution for the weights: $w_i = a_i^2 s_i$, $s_i \sim \text{Bernoulli}(\pi_i)$, $a_i^2 \sim p(a_i^2)$. The "spike" is represented by $\text{Bernoulli}(\pi_i)$, and determines whether or not the candidate kernel is included in the composition. The "slab" is represented by $p(a_i^2)$ and is a prior on the weights themselves (note that this is equivalent to a prior on the kernel's amplitude hyperparameter). Our experiment settings for the DP parameters are given in appendix 10.2.5.

For *sparsity*, the spike-and-slab distribution encourages $w_i$ to be zero, which induces sparsity in the total number of candidate kernels included. Smaller values of $\pi_i$ encourage greater sparsity (since $w_i = 0$ if $s_i = 0$). For *stability*, when $G_{\text{parent}}$ is conditioned on $K_{\text{parent}}$, it places a higher probability of including the same candidate kernels as $K_{\text{parent}}$. Specifically, if candidate kernel $j$ was in the previous composition, then we set its probability $\pi_j$ close to 1. Since the same candidate kernels are more likely to be included from one time step to the next, the evolutions will be more stable.

**Modeling online data generated by kernel evolutions.** Now that we have defined the evolution of a composition between time steps, we are prepared to define how $\mathcal{D}_{u,t}$ is generated as a result of such

evolutions:

$$K_{u,t} \sim H_{K_{u,t-1}} \tag{4}$$

$$\sigma_{u,t}^2 \sim p(\sigma_{u,t}^2 | K_{u,t-1}) \tag{5}$$

$$\theta_{u,t} \sim p(\theta_{u,t} | K_{u,t}) \tag{6}$$

$$f_{u,t} \sim GP(0, K_{\theta_{u,t}}) \tag{7}$$

$$Y_{u,t} \sim \mathcal{N}(f_{u,t}, \sigma_{u,t}^2). \tag{8}$$

**Eq. 4: The kernel composition at time step $t$ is sampled from the distribution over compositions defined by the preceding composition at $t-1$.** The DP from eq. (3) allows us to transfer information about the likeliest next composition (evolution) between users with the same parent composition. As discussed, the base distribution of the DP also encourages stability; if a user has a "linear" kernel one time step, they are likely to build from the "linear" kernel at the next time step.

**Eq. 5 and 6: The prior over hyperparameters depends on the composition.** If a composition increases in complexity from one time step to the next, then the corresponding observation noise should decrease, since the data-generating function is stationary. To encode this behavior, our prior on the noise $\sigma_{u,t}^2$ is a decreasing function of the number of components. Our prior over the remaining kernel hyperparameters helps separate the model classes. For example, a squared-exponential kernel with a large lengthscale maps to functions that behave linearly; a prior on the lengthscale would minimize overlap between the Linear and SE kernels. See appendix 10.2.5 for the priors used in experiments.

**Eq. 7 and eq. (8): Each user's function is sampled from a Gaussian Process.** The GP is defined by the kernel and the final observations are corrupted by Gaussian noise. Though in our model $Y_{u,t}$ will overlap in data with $Y_{u,t-1}$, we use different generative processes for each by noting that the bias–variance trade-off, and therefore the best composition, may differ between the two overlapping data sets.

### 5.3 Pilot Training: Offline Inference for KEM

We are now prepared to infer the set of kernel evolutions that generated the pilot user's "online" data. We represent each DP as a Chinese Restaurant Process (CRP) to track the assignments of pilot data to clusters. We will describe inference for the parameters *of a single CRP* (conditioned on a $K_{\text{parent}}$ composition). This inference jointly occurs for each unique $K_{\text{parent}}$ composition.

We must infer the set of $K_{\text{child}}$ clusters and the assignments of pilot datasets (datasets for each user, at each time step) to these clusters. The unknown parameters are $\boldsymbol{\Theta} = (\boldsymbol{Z}, \boldsymbol{K}, \boldsymbol{\theta}, \boldsymbol{\sigma^2})$. Assuming there are $N$ datasets and $C$ clusters in the $K_{\text{parent}}$-th CRP, the cluster assignment of the $n$-th dataset is given by $\boldsymbol{Z}_n \in \{1, \ldots, C\}$. Each cluster represents a kernel. The $c$-th kernel is defined a composition $\boldsymbol{K}_c$, hyperparameters $\boldsymbol{\theta}_c$, and observation noise $\boldsymbol{\sigma_c^2}$. Inference of the unknown parameters involves sampling from the joint posterior, $p(\boldsymbol{Z}, \boldsymbol{K}, \boldsymbol{\theta}, \boldsymbol{\sigma^2} | \mathcal{D})$, where $\mathcal{D} = \{\mathcal{D}_{u,t}\}$ is the set of all training user's data (separated by time step $t$ so that it is processed as though it is "online"). Each posterior sample results in a selection model like the one shown in fig. 2. Following the marginal Gibbs sampler from Gelman et al. (2013), we alternate between the following two steps (with further detail in appendix 10.3):

**1. Assigning datasets to clusters.** The goal is to sample from the posterior $p(\boldsymbol{Z} | \boldsymbol{K}, \boldsymbol{\theta}, \boldsymbol{\sigma^2}, \mathcal{D})$. For a single dataset $\mathcal{D}_{u,t}$, we sample from a multinomial posterior defined by the likelihood that the kernel within the cluster generated $\mathcal{D}_{u,t}$ and the prior probability the cluster assignment in eq. (1).

**2. Assigning kernels to clusters.** The goal is to sample from the posterior $p(\boldsymbol{K}, \boldsymbol{\theta}, \boldsymbol{\sigma^2} | \boldsymbol{Z}, \mathcal{D})$. Each dataset in the cluster is treated as a *sample* from a GP with the same kernel, and we use this data to select a kernel to represent the cluster. It is possible for two different clusters to have the same composition but different hyperparameters. We obtain samples from the posterior distribution of the kernel via the Metropolis-Hastings algorithm (appendix 10.3.2).

### 5.4 Deployment testing: Using KEM for New Users Online

Up to this point, we have described how to infer a set of kernel evolutions defined by the generative model section 5.2 by applying the inference procedure in section 5.3 on pilot data. In algorithm 1, we define a selection model that leverages these learned evolutions to select a kernel for a new test user $u^*$ at time $t$.

---

**Algorithm 1** Selection method for KEM

---

**Input**    :  Test user's current data $\mathcal{D}_{u^*,t} = \{\mathbf{X}_{u^*,t}, \mathbf{y}_{u^*,t}\}$, preceding composition $K_{u^*,t-1}$
**Output :** kernel with composition $K^*$ and hyperparameters $\hat{\theta}_{K^*}$
Define set of potential compositions $\boldsymbol{K^*} = \{\boldsymbol{K}|K_{u^*,t-1}\} \cup \{K_{u^*,t-1}\}$
**for** $K \in \boldsymbol{K^*}$ **do**
  Initialize hyperparameters $\theta_K$ to pilot user's hyperparameters
  Optimize $\hat{\theta}_K = \max\limits_{\theta_K} \log p(\mathbf{y}_{u*,t}|\mathbf{X}_{u^*,t}, \theta_K)$
**end**
Select $K^* = \min\limits_{K \in \boldsymbol{K^*}} \mathrm{BIC}(K; \hat{\theta}_K)$
Return $K^*, \hat{\theta}_{K^*}$

---

Our method minimizes the Bayesian Information Criterion (BIC): $BIC(K_\theta) = |\theta|\ln(n) - 2\ln(L_{\hat{\theta}})$, where $L_{\hat{\theta}}$ is the log marginal likelihood of the data, for a GP with composition $K$ and MLE hyperparameters $\hat{\theta}$. The BIC is a common model selection metric that penalizes hyperparameter complexity (Duvenaud et al., 2013; Kim & Teh, 2018); though it can be replaced with *any* model selection metric, KEM is still needed *in addition*, to restrict the set of compositions to those that are stable and sparse for online selection. KEM restricts $\boldsymbol{K^*}$, the set of compositions considered, to those that were identified for pilot users with the *same preceding composition*. We also allow *no evolution* to occur, by including the previous composition $K_{u^*,t-1}$ in the set. When $t = 1$, $\boldsymbol{K^*}$ is initialized to those that were found at $t = 1$ for pilot users. Conceptually, our approach corresponds to "traversing" the graph in fig. 2 to select a kernel; the test user is currently at node $K_{u^*,t-1}$ and $\boldsymbol{K^*}$ contains the children of the current node.

## 6 Experimental Setup

We evaluate a method's ability to select compositions that are *sparse* (includes only relevant features), *stable* (the same features are consistently included across time steps), and of *adaptive complexity* (the kernels perform well across different data sizes). Furthermore, any method for online deployment must be sufficiently *scalable* to multiple users and time steps. All empirical experiments are over 10 independent trials.

### 6.1 Baselines

KEM has three essential aspects, and each of our baselines omits at least one of these aspects: KEM learns a model selection strategy from pilot users (pilot training), learns the kernel compositions at different time points (adaptive complexity), and models composition evolutions between time points (stable). The **Memoryless** method (as in (Tong et al., 2021), but with no variational GP approximations) does not have a pilot training phase and always requires a new round of selection in deployment. The **ARD** method represents using a GP with a fixed kernel composition– the ARD kernel– throughout the deployment, with no kernel re-selection. The ARD kernel is a composition of SE kernels applied to each dimension $d$ of a $D$ dimensional data set (Rasmussen & Williams, 2006): $\prod\limits_{d=1}^{D} \mathrm{SE}_d$. Note that the lengthscales of ARD must still be optimized to new data. The **Final** (Titsias & Lázaro-Gredilla, 2011) method transfers knowledge obtained on the *final data set* of pilot users, ignoring the potential to overfit when there are fewer data points for the test user. Finally, **Stratified** is an ablation of our approach that only transfers kernels from pilot to test users if they were found at the same time step. It still has our innovation of suggesting kernels based on how much data has been collected but does not enforce stability, since it treats kernels at subsequent time steps independently. Overall, the methods with pilot training (Final, Stratified, KEM) transfer the *kernel compositions*, but the *kernel hyperparameters* are re-optimized to the new user's data during selection.

## 6.2 Candidate Kernels

The set of candidate kernels contains "atomic kernels," which are combinations of three commonly-used kernel functions (Tong et al., 2021; Duvenaud et al., 2013) – linear (LIN), periodic (PER), and squared-exponential (SE) – applied to a single dimension of the data. For example, LIN0 denotes a linear kernel function that operates on the 0-th dimension of the data. WHITE means no kernel is selected. The candidate kernel set also includes first-order interactions between these "atomic kernels". For example LIN1 × LIN1 or LIN1 × SE1. Finally, the ARD kernel is included, so that the selection methods are comparable to the SE-ARD method.

## 6.3 Overview of Datasets

| Data set | # instances | # features | # users | # instances/user | # time steps |
|---|---|---|---|---|---|
| Synthetic | 1800 | 1 | 60 (10 train/50 test) | 30 | 6 |
| UCI: Energy | 768 | 8 | 15 (7 train/8 test) | 50 | 10 |
| UCI: Housing | 506 | 13 | 10 (5 train/5 test) | 50 | 10 |
| UCI: Fires | 244 | 15 | 8 (4 train/4 test) | 30 | 10 |
| UCI: Concrete | 1030 | 8 | 20 (10 train/10 test) | 50 | 10 |
| MIMIC-III | 975 | 14 | 16 (8 train/8 test) | varies | varies |
| HeartSteps | 1535 | 23 | 37 (18 train/19 test) | varies | varies |

Table 1: Descriptions of datasets. For MIMIC and HeartSteps, different users had different available data sizes, due to missingness in the data or different lengths of stay in the ICU.

The datasets used in our experiments, reflected in table 1, have different properties. By design, the users in the UCI data sets are *homogeneous*, because they were artificially constructed by randomly splitting the data. The MIMIC, HeartSteps, and Synthetic data are expected to have more heterogeneity– "clusters" of users for whom different compositions are more appropriate. In terms of *noise*, the Synthetic (by design) and UCI: Fires (a simpler prediction task) data are relatively low noise compared to the remaining data sets. We are particularly interested in the performance of the kernel selection methods on the high noise, high heterogeneity data– HeartSteps and MIMIC-III– since these characteristics reflect our health setting.

# 7 Results

In section 7.1, we begin with a pedagogical comparison of the methods on synthetic data and then demonstrate that the same takeaways hold on real data sets in section 7.2.

## 7.1 Demonstrative Results on Synthetic Data

We will begin by examining what the selection models learned during pilot training and link them to performance on selecting kernels for new users in testing. We find that KEM's ability to identify simpler, intermediary compositions during pilot training is crucial to performance in low-data regimes.

**Synthetic data.** We constructed a 1-D data set that contains two "types" of users. For half of the users, we sampled each function from a GP prior with a LIN0 + PER0 kernel. The other half's functions were sampled from a SE0 kernel. Each user's data arrives in batches of 5 data points over 6 total time steps (for a total of 30 data points in the end). For a test user $u^*$, the train set is the cumulative data $\mathcal{D}_{u^*,t}$ observed on the user up to time $t$. The test set is composed of 200 uniformly spaced points along the x-axis from $[0, 20]$.

**Sparsity and stability: KEM tells a cohesive story about how kernels evolve with more data.** In fig. 3 we show an example of the kernel compositions learned by the Final, Stratified, and KEM methods during pilot training; the ARD and Memoryless methods do not have a pilot phase. As expected, Stratified and KEM learn simpler, intermediary kernels to account for varying time steps, while Final does not. However, Stratified chooses kernels that lack congruence over time, since it does not share information about

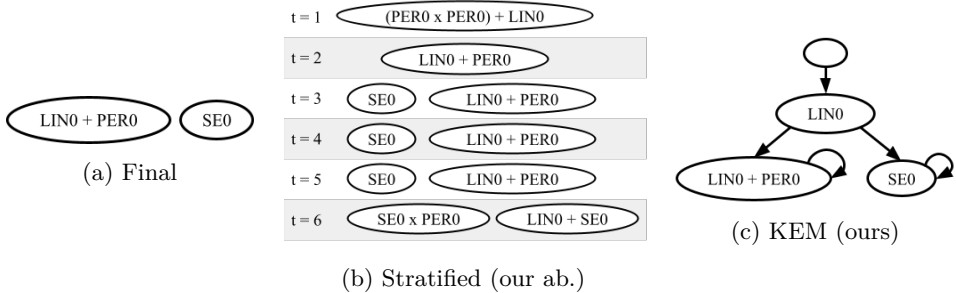

(a) Final

(b) Stratified (our ab.)

(c) KEM (ours)

Figure 3: **During pilot training, KEM learns evolutions in which simple kernels consistently precede complex kernels, whereas baselines do not.** Final identifies two clusters, the two final compositions (as expected). Stratified randomly adds and drops components, such as when it goes from a $PER0 \times PER0 + LIN0$ composition to a $LIN0 + PER0$ composition from $t = 1$ to $t = 2$.

the composition across time steps. On the other hand, KEM tells a clear story about how the data is initially best fit by a LIN0 kernel and subsequently evolves into either a LIN0+PER0 or SE0 kernel (the true kernels). These relationships were enforced by our model's definition of an "evolution."

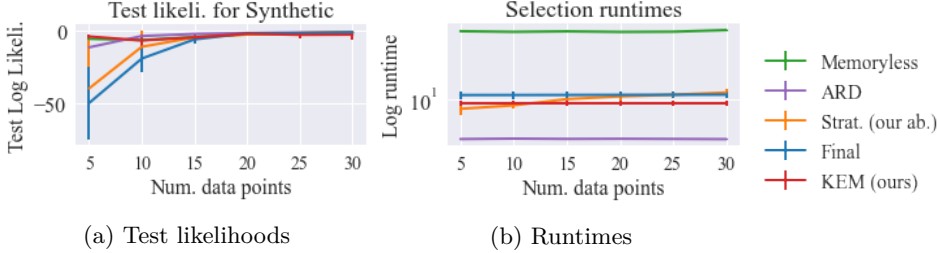

(a) Test likelihoods

(b) Runtimes

Figure 4: **KEM quickly selects high-performing kernels (log-likelihood) for new users across varying amounts of synthetic data, by successfully managing the bias-variance trade-off.** Left: log-likelihood. Right: runtime in log scale. Error bars: two standard deviations. Memoryless is included, but overlapping, with KEM.

**Feasibility: The pre-trained selection methods select kernel compositions at a rate that is feasible for online learning.** In fig. 4b the methods with a pilot training phase are orders of magnitude faster at selecting a kernel than Memoryless, which must re-perform kernel selection for each new user and time step. ARD is quickest because it performs no selection. Though one might update the kernel on a weekly or monthly time scale in a health setting, the runtime required for Memoryless would grow rapidly with the number of users. Alternatively, the computation cost for the pre-trained methods is primarily spent optimizing the hyperparameters for a small set of kernels (on the order of 2 or 3 comparisons for new data).

**Adaptive complexity: KEM manages the bias-variance trade-off by regularizing the kernel composition in low data regimes.** We expect a method that manages the bias-variance trade-off to have a high test likelihood when there is little data and after more data has been observed. In fig. 4, we see evidence of KEM managing the bias-variance trade-off by maintaining the best test performance across time steps. The mechanism through which KEM manages this trade-off is by regularizing the kernel composition when there is little data. In fig. 5, KEM initially selects simpler compositions– LIN0 and White– for new users. KEM eventually selects more complicated compositions– LIN0 + PER0, LIN0 + SE0, and SE0– as necessitated by the test user's data. KEM's use of simpler compositions when data size is low reduces the chances of overfitting the hyperparameters. Memoryless requires more data from the test user to identify the true composition and underfits the data.

Figure 5: **KEM chooses simpler kernels where there are** 5 **data points and more complex kernels (including ground truth) when there are** 30 **data points.** We display distributions over 5 most common compositions selected across all 10 trials when the ground truth is LIN0 + PER0. ARD method is SE0 kernel by default.

On the other hand, we see evidence that the more complicated compositions initially selected by the baseline methods (e.g. a LIN0 + PER0 kernel) lead to overfitting hyperparameters (fig. 9). Fig. 4a demonstrates that overfitting in low data regimes results in poor test likelihoods.

## 7.2 Generalization to Real Data

In this section, we verify that the behaviors of the selection methods from section 7.1 regarding sparsity, stability, and adaptive complexity hold on more complex data. We omitted further experiments with Memoryless due to impracticality with respect to computation time in online settings. For a test user $u^*$, the train set is the cumulative data $\mathcal{D}_{u^*,t}$ observed on the user up to time $t$. The test set is the incoming data from the next time step: $\mathcal{D}_{u^*,t+1} \setminus \mathcal{D}_{u^*,t}$. This procedure mimics the way that models are used in practice; a kernel is selected with the user's cumulative data and then used to make predictions on incoming data, until the next time step, when the kernel must be selected again.

**UCI data.** We include four UCI regression tasks: Energy (Tsanas & Xifara, 2012), Concrete (Yeh, 1998), Boston Housing (Harrison Jr & Rubinfeld, 1978), and Fires (Abid & Izeboudjen, 2019). We randomly split these batch, single-task datasets into "users" and further into "batches" of 5 points each to reflect our setting.

**MIMIC-III: Predicting plateau pressure.** We use the Medical Information Mart for Intensive Care (MIMIC-III) data set Johnson et al. (2016), which contains patient's physiological readings from the ICU. The regression goal is to predict plateau pressure– the amount of pressure applied to the airways during mechanical ventilation– from the patient's other vitals (e.g. heart rate). We grouped patients into 16 tasks via their *diagnosis upon admission*, such as sepsis, and assumed data arrive in batches of 10 points.

**mHealth: Imputing missing wearable data.** HeartSteps V1 (Klasnja et al., 2019) was an mHealth study that helped sedentary individuals increase physical activity (i.e. number of active minutes) through contextually-tailored interventions. In mHealth, a major source of missing data is when users forget to wear their tracking devices (Seewald et al., 2019). Our goal is to actively impute each participant's missing values of "daily active minutes" (square-root transformed), from other data sources (e.g. location data). We assume that the data from 37 users arrive in batches of 5 points.

**Adaptive complexity.** We expect KEM to have high test-likelihood for varying levels of data. In fig. 6, patterns from the synthetic experiments hold; KEM maintains high likelihood, while Final consistently overfits when there are small amounts of data. We believe ARD performs worse on real data because it has more opportunity to overfit the lengthscales to the multi-dimensional datasets. All methods perform similarly on Fires, a simpler regression task with low noise.

**Stability.** On synthetic data, we noticed Stratified chose kernels that are inconsistent across time steps, and we demonstrate that this behavior is exacerbated on real data in table 2.

**Sparsity.** In fig. 7, we provide a basic interpretation of the evolutions that KEM learned on the health data. This is **not** to replace a domain expert but to demonstrate that KEM selects sparse compositions that are easy to examine. In contrast, the composition implied by ARD is not sparse and is difficult to use in relating the features to the predictions.

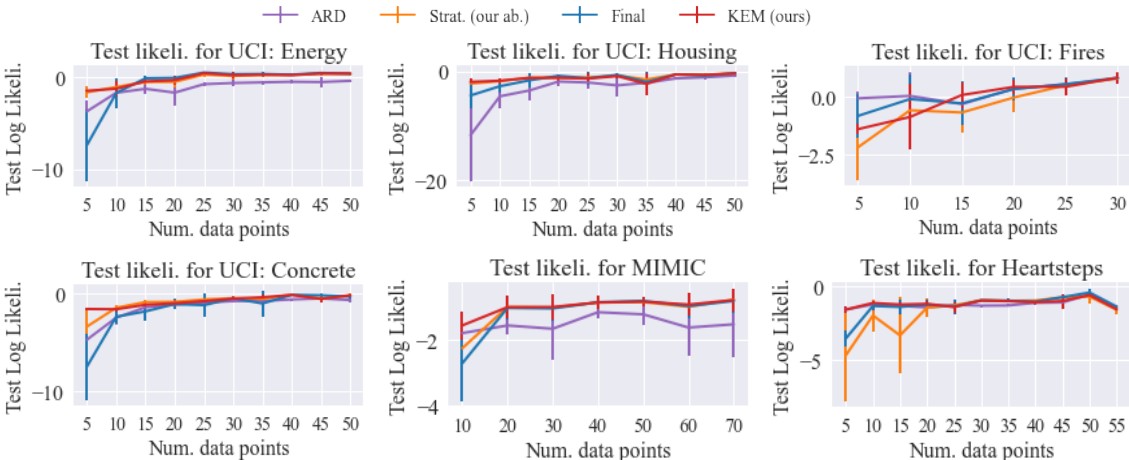

Figure 6: **KEM is the only kernel selection method to consistently perform well for all levels of data in 5 of 6 datasets (including real health settings).** All methods perform similarly on the simplest of the data sets (fires). Error bars are 95% confidence intervals.

| Dataset | UCI: Energy | UCI: Housing | UCI: Fires | UCI:Concrete | MIMIC | HeartSteps |
|---|---|---|---|---|---|---|
| KEM (ours) | $\mathbf{0.08} \pm 0.05$ | $\mathbf{0.07} \pm 0.04$ | $0.29 \pm 0.17$ | $\mathbf{0.06} \pm 0.04$ | $\mathbf{0.07} \pm 0.1$ | $\mathbf{0.22} \pm 0.05$ |
| Strat. (our ab.) | $1.09 \pm 0.14$ | $0.73 \pm 0.3$ | $0.56 \pm 0.2$ | $0.46 \pm 0.14$ | $0.22 \pm 0.12$ | $0.62 \pm 0.15$ |

Table 2: **KEM consistently includes the same features across time steps.** Table shows avg. # of features included in kernel at one time step but dropped the next for each test user (lower is better). We report 95% confidence intervals.

For the MIMIC application of predicting plateau pressure, KEM prioritizes the "peak inspiratory pressure" feature earlier in the evolution. Plateau pressure is directly related to the peak pressure through the resistance of airflow in the lungs. Features farther down the evolution, such as inspired oxygen and respiratory rate, also relate to passive and mechanical ventilation (Hagberg & Fasa, 2022). For imputing missing wearable values in HeartSteps, we expected "GoogleFit (daily) steps" to be the most predictive feature. Interestingly, some evolutions lead to a linear kernel on the "day in study," which we hypothesize is because some users decrease activity throughout the course of a study; this trend may only be apparent as enough days (and data points) are observed for a user.

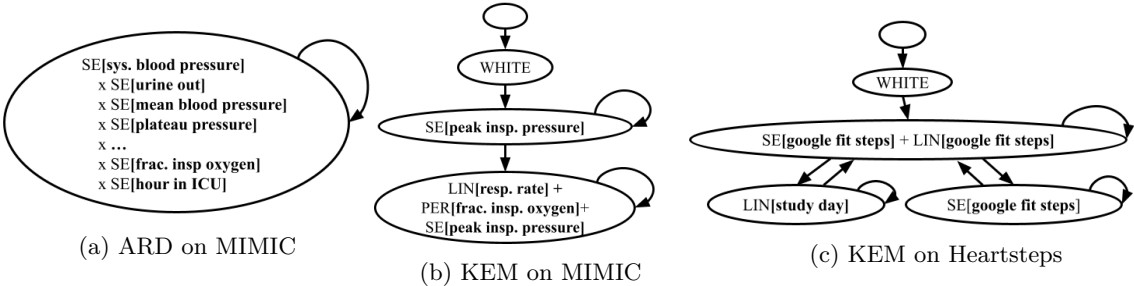

(a) ARD on MIMIC      (b) KEM on MIMIC      (c) KEM on Heartsteps

Figure 7: **KEM selects sparse kernel compositions that are easily examined.** Examples of the selection model learned by KEM on MIMIC and HeartSteps shown. ARD "selects" a composition formed by multiplying an SE kernel for each feature; remaining 8 of 14 features omitted due to space.

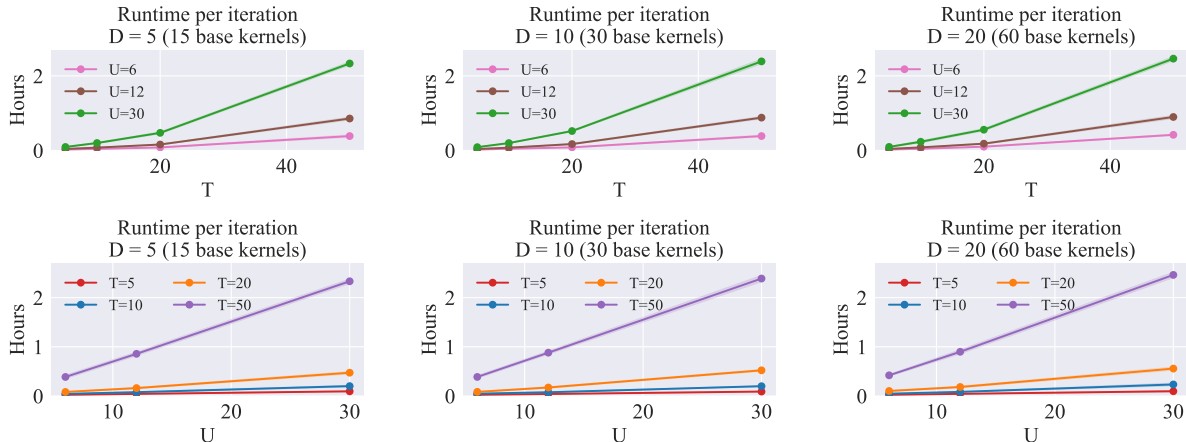

Figure 8: **KEM train times are most affected by the number of time steps.** In top row, KEM scales exponentially for increasing $T$. In bottom row, KEM scales linearly for increasing $M$. The runtime per iteration does not scale with number of dimensions (each column).

### 7.3 Scalability of KEM in pilot training

Our experiments have demonstrated the benefits of using KEM *during online deployment* to select sparse and adaptive kernel compositions. Though by design, KEM achieves these benefits online by frontloading the computational cost of kernel selection to a long pilot training phase, for practical consideration, we explore how pilot training scales with the size of the pilot data.

**Experimental outline.** Recall that our pilot training for KEM involves Gibbs sampling. The dependent variable is the *runtime per sample* drawn from the posterior. The independent variables relate to the size of the pilot data and include the number of users $U$, the number of time steps $T$, and the number of input dimensions $D$. We constructed the pilot data as a $D$-dimensional data set that contains three "types" of users– a LIN0 + PER0 composition, a LIN0 + PER1 composition, and a SE1 composition– and there are the same number of users per type. Each user's data arrives in batches of 5 data points over $T$ total time steps. We train on each pilot dataset for 100 posterior samples, though in practice the total number of samples required for convergence will vary. The reported runtimes are on a 4 core Intel "Cascade Lake" CPU.

**KEM training scales better to more users (linearly) than to more time steps (exponentially).** Figure 8 indicates that in general, KEM is most suitable for applications where the *pilot data* has a large number of users but a small number of time steps. In our experiments on performance during deployment, we demonstrated that KEM is useful for managing the bias-variance trade-off in earlier time steps. Therefore, even when trained on pilot data with small $T$, KEM can achieve its purpose of appropriately adapting the complexity of the kernel early on. Note that it is also possible to form the time steps in the pilot data with irregular batch sizes (that is, smaller batch sizes initially, and then larger batch sizes later on).

**The runtime per posterior sample is not affected by the dimensionality of the training data.** This is due to the fact that our sampling scheme searches over kernel compositions by randomly sampling subsets of the base kernels. These operations do not scale with the size of the base kernel set; however, a larger number of base kernels will make this sampling-based search more difficult, and therefore, require a larger number of samples overall.

## 8 Discussion

**Limitations and Future Work**

*Scalability of KEM in pilot training.* KEM scales poorly with more pilot users and time steps, since we require repeated GP marginal likelihood per user and time step. However, in our setting, this is a manageable *one-time cost* incurred during the long pilot period. Our only goal for pilot training is that it enables us to perform online kernel selection for a large number of users in the real deployment (e.g. in fig. 4b, only 10 seconds per time step, on average, is required). In future work, we are excited to incorporate advancements in scalable GPs, such as (Kim & Teh, 2018), to reduce the cost of the marginal likelihood calculations.

*Limited interaction terms.* The number of interaction terms considered in this approach is restricted to those in the candidate kernel set. This is acceptable in health, where we do not expect to find higher-order effects due to noise and sparse data (Trella et al., 2022). In future work, we may consider actively adapting the candidate set to include more interactions.

*Rigidity of the kernel evolutions from pilot to test.* Though KEM's strategy, which limits the space of kernels considered for test users, is exactly what allows kernel selection to occur scalably in our setting, it also means that test users are limited to the compositions found on training users. An interesting direction for future work is to use the set of kernel evolutions from pilot training as a *prior* for new users, rather than a hard constraint. The task of adapting the kernel evolution set online remains a challenge.

**Conclusion** In this work, we presented a method for compositional kernel selection in the online, multitask setting. We illustrated that learning "evolutions" of kernel compositions is most beneficial in the low-volume and/or heterogeneous user regime, when overfitting poses a significant concern. We demonstrated across a variety of data sets, including two health applications, that our approach quickly selects kernels that outperform baselines in predictive performance, sparsity, and stability – all crucial considerations for real-world deployment.

## 9    Acknowledgements

This material is based upon work supported by the National Science Foundation under Grant No. IIS-1750358 and the Graduate Research Fellowship Program under Grant No. DGE 2140743. Any opinions, findings, and conclusions or recommendations expressed in this material are those of the author(s) and do not necessarily reflect the views of the National Science Foundation. The research reported in this publication was supported by the National Institute of Biomedical Imaging and Bioengineering of the National Institutes of Health under award number OD P41EB028242. ES's work on the project was supported by a gift fund from Benshi.ai.

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

## 10  Appendix

### 10.1  Additional Results

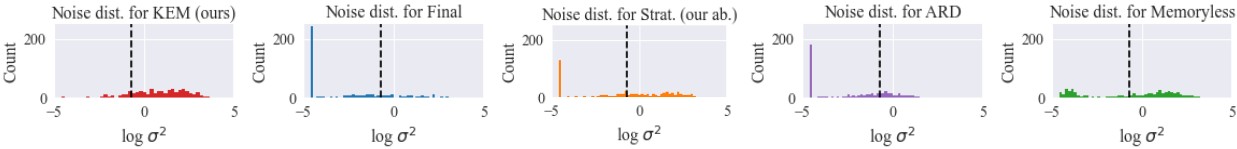

Figure 9: **Baseline methods frequently underestimate the observation noise hyperparameter, a sign of overfitting.** We plot the log-distribution of $\sigma^2$ of test users with 5 training points. True $\sigma^2 = 0.5$ (dotted black line). We constrain $\sigma^2 \geq 0.01$ during optimization for numerical stability (hence the spikes around $-5$).

### 10.2  Experimental Details

#### 10.2.1  Datasets

**Generation of synthetic data.** Half of the user's functions were generated from a LIN0 + PER0 kernel composition. The other half's functions were generated from a SE0 kernel. To generate the synthetic data for each user, we first sample the hyperparameters of the ground-truth kernel. The users with a LIN0+PER0 compostion had hyperparameters that were sampled from as follows:

- $\theta_{\text{lengthscale}} \sim \text{Log-Normal}(0, 0.1)$

- $\theta_{\text{period}} \sim \text{Log-Normal}(1.386, 0.1)$

- $\theta_{\text{amplitude}} \sim \text{Log-Normal}(1.609, 0.1)$

- $\theta_{\text{shift}} \sim \text{Normal}(5, 1)$

The users with a SE0 kernel had hyperparameters that were sampled as follows:

- $\theta_{\text{lengthscale}} \sim \text{Log-Normal}(0, 0.1)$

- $\theta_{\text{amplitude}} \sim \text{Log-Normal}(2.302, 0.1)$

The observation-noise for the synthetic experiments was fixed for all users as $\theta_{\text{noise}} = 0.5$. After defining the user's kernel– by assigning the composition and hyperparameters– we sample the user's function from a GP prior with the kernel.

**Pre-processing of real data.** For the real data experiments, features are scaled to the range of $[0, 1]$ using min-max normalization. The prediction targets are standardized ($y' = \frac{y - \text{mean}(y)}{\text{std}(y)}$). The prediction targets for each of these data sets are as follows:

- UCI Energy: heating load

- UCI Housing: housing price

- UCI Fires: Build up Index (BUI) for Bejaia data, Initial Spread Index (ISI) for Sidi data.

- UCI Concrete: concrete compressive strength

- MIMIC-III: plateau pressure

- HeartSteps: square-root transformed daily values of "number of active minutes."

**MIMIC-III Pre-processing** The pre-processing steps for MIMIC are given below:

- Original number of data points: 331532 points

- Select data points where user is on ventilation: 39125 points

- Remove data points outside a valid range (as defined by medical expert): 22046 points

- Remove data points missing the prediction target (plateau pressure): 3321 points

- Group data points by diagnosis upon admission. Remove diagnoses with fewer than 30 points per diagnosis: 975 points

**HeartSteps Pre-processing** The pre-processing steps for HeartSteps are as follows:

- Original number of data points, aggregated across all users: 1658 points

- Take square-root transformation of the target value (daily active minutes): 1658 points

- Drop points where daily active minutes are zero: 1535 points

### 10.2.2 Train/Test Splits

In the synthetic experiments, we have access to the ground-truth kernel which can be used to evaluate the quality of the kernels selected for test users. This is not the case for real data. As a result, the training and testing data splits differed slightly between the experiments involving real and synthetic data.

**Synthetic Experiments**

- Training Users (10 training users)

- Testing Users (50 test users)

  – Training data is used to select the kernel. The training data is the cumulative data set at each time step.
  – Testing data is used to evaluate the kernel. Since we have access to the ground-truth kernel, the test data is the function evaluated at 200 uniformly spaced points along the X-axis with noise.

**Real Data Experiments** For the real data experiments, users were randomly assigned to either the training or testing set in a 50 : 50 split.

- Training Users (50% of total users randomly selected)

- Testing Users (remaining 50% of users)

  – Training data is used to select the kernel. The training data is the cumulative data set at each time step.
  – Testing data is used to evaluate the kernel. Since we do not have the ground truth kernel, the test data refers to the future (unobserved) data that will arrive at the next time step.

The train/test split for the test users mirrors the realistic process of selecting a model for a user based on his cumulative data and then using the selected model to make predictions on the data observed until the next update point.

### 10.2.3 Candidate Kernel Pool

**Kernel functions**   Consistent with the previous kernel selection literature, we consider the linear, periodic, and squared-exponential kernel functions.

$$k_{\text{SE}}(x, x') = a^2 \exp\left(-\frac{(x - x')^2}{2\ell^2}\right) \tag{9}$$

$$k_{\text{Per}}(x, x') = a^2 \exp\left(-\frac{2\sin^2(\pi|x - x'|/p)}{\ell^2}\right) \tag{10}$$

$$k_{\text{Lin}}(x, x') = a^2(x - c)(x' - c) \tag{11}$$

$$k_{\text{White}}(x, x') = a^2\delta_{i,j} \ , \delta_{i,j} = 1 \text{ if } x_i = x_j \tag{12}$$

The squared-exponential kernel (eq. (9)) has two hyperparameters, the lengthscale $\ell$ and amplitude $a^2$. The periodic kernel (eq. (10)) has three hyperparameters, the lengthscale $\ell$, amplitude $a^2$, and period $p$. The linear kernel (eq. (11)) has two hyperparameters, the amplitude $a^2$ and offset $c$. Finally the white kernel (eq. (12)) has one hyperparameter, the amplitude or noise parameter, $a^2$.

**Data specific Kernel pools**   The basic candidate kernel pool is formed by pairing each kernel function with each dimension of the data. The synthetic experiments include first-order interaction terms between these candidate kernels. The real data experiments include interaction terms on features that are identified as important to the prediction target in the literature (for example, whether or not it is a weekend is an important feature in determining how active a user will be for that day). Finally, the ARD kernel (the product of an SE kernel on each dimension of the data) is included in the candidate pool as well, so that our selection method can be directly compared to the SE-ARD method.

- Synthetic experiments: basic kernel functions on the first (and only) dimension of the data, along with interaction kernels up to the second degree: $\{\text{LIN0}, \text{PER0}, \text{SE0}, \text{LIN0} \times \text{PER0}, \text{LIN0} \times \text{SE0}, \text{LIN0} \times \text{LIN0}, \text{PER0} \times \text{SE0}, \text{PER0} \times \text{SE0}\}$

- Real data experiments:
    - UCI Energy: 16 total candidate kernels.
    - UCI Housing: 38 total candidate kernels.
    - UCI Fires: Interaction term on windspeed. 35 total candidate kernels.
    - UCI Concrete: Interaction term on cement and age. 39 total candidate kernels.
    - MIMIC-III: Interaction term on whether patient is on vasos (binary). 50 total candidate kernels.
    - HeartSteps: Interaction term on whether user is "traveling" (binary) and whether or not it is a "weekend" (binary). 95 total candidate kernels.

### 10.2.4 KEM Inference Parameters

- Gibbs sampling during pilot training:
    - until convergence, up to 200 iterations
    - 10 iterations of global updates (assigning datasets to clusters)
    - 100 samples from MH sampling algorithm during local updates (assigning kernels to clusters)

- MH sampling algorithm:
    - Proposal distribution for kernel hyperparameters: $\mathcal{N}(0, \sigma_{\text{hyper}})$ where $\sigma_{\text{hyper}} = 0.1$ for synthetic experiments and $\sigma_{\text{hyper}} = 0.05$ for real data experiments (because features are normalized)

### 10.2.5   KEM Priors

**Dirichlet Process**

- $\alpha = 1$

- For the base distribution $G_{\text{parent}}|K_{\text{parent}}$:

  - For $s \sim \text{Bernoulli}(\pi)$
    * $\pi_i = 0.1$ if $K_i$ is an atomic candidate kernel and $K_i$ not in $K_{\text{parent}}$
    * $\pi_i = 0.02$ if $K_i$ is an interaction candidate kernel and $K_i$ not in $K_{\text{parent}}$
    * $\pi_j = 0.9$ for $K_j$ in $K_{\text{parent}}$
  - For $a_i^2 \sim p(a_i^2)$: Because $a_i^2$ determines how the candidate kernel is scaled, it also refers to the common "amplitude" hyperparameter included in most kernel definitions (see section 10.2.3); therefore, $p(a_i^2)$ is a prior over the amplitude hyperparameter defined in the next paragraph.

**Kernel Hyperparameter Priors**   The prior distribution on the kernel hyperparameters are as follows:

- Lengthscale: $\log p(\theta_{\text{lengthscale}}) = \mathcal{N}(0, 2)$ for synthetic data, $\log p(\theta_{\text{lengthscale}}) = \mathcal{N}(0.2, 0.5)$ for real data

- Period: $\log p(\theta_{\text{period}}) = \mathcal{N}(5, 0.25)$ for synthetic data, $\log p(\theta_{\text{period}}) = \mathcal{N}(0.2, 0.25)$ for real data

- Amplitude: $\log p(\theta_{\text{amplitude}}) = \mathcal{N}(0, 2)$

- Observation noise: $\log p(\theta_{\text{noise}}) = \mathcal{N}(0, 2)$

- Shift: $p(\theta_{\text{shift}}) = \mathcal{N}(0, 0.1)$

The lengthscale and period priors differ between the synthetic and real experiments because the features are normalized between $[0, 1]$ for the real experiments. For the synthetic experiments, the choice of $\mu = 5$ for the prior on $\theta_{\text{period}}$ reflects our general estimate that the period may be around $\frac{1}{4}$ the domain of the input space.

**Prior on the observation noise for the KEM.**   The KEM (ours) model enforces a prior on the observation noise which depends on the parent composition. Let $N_{\text{parent}}$ denote the number of kernel components in the parent composition:

$$\log p(\theta_{\text{noise}}|K_{\text{parent}}) = \left\{ \begin{array}{ll} \mathcal{N}(2, 0.5), & \text{for } N_{\text{parent}} = 0 \\ \mathcal{N}(1, 1), & \text{for } N_{\text{parent}} = 1 \\ \mathcal{N}(0, 2), & \text{for } N_{\text{parent}} > 1 \end{array} \right\}$$

### 10.3   Inference Details for Kernel Evolution Model

### 10.3.1   Details of Gibbs sampler steps

**Assigning datasets to clusters.**   The seating assignment for the $m^{\text{th}}$ customer is sampled from a multinomial posterior with the probability:

$$p(Z_n = c|\mathcal{D}, \boldsymbol{K}, \boldsymbol{\theta}, \boldsymbol{\sigma^2}) \propto p(\mathbf{y}_n|\mathbf{X}_n, \boldsymbol{K}_c^-, \boldsymbol{\theta}_c^-, \boldsymbol{\sigma}_c^{2-}, Z_n = c)p(Z_n = c|\mathbf{Z}^-) \tag{13}$$

where $\boldsymbol{\mathcal{M}}^-$, $\boldsymbol{\theta}^-$, $\boldsymbol{\sigma}^{2-}$, and $\mathbf{Z}^-$ are the remaining kernel compositions, hyperparameters, noise levels, and seating assignments at occupied tables after unseating customer $m$. Here $p(\mathbf{y}_n|\mathbf{X}_n, \boldsymbol{K}_c^-, \boldsymbol{\theta}_c^-, \boldsymbol{\sigma}_c^{2-})$ is the model likelihood. The table asssignment probability $p(Z_n = c|\mathbf{Z}^-)$ is as determined by a Chinese Restaurant Process.

**Assigning kernels to cluster.** We assign a kernel to a cluster $c$ by sampling from the posterior distribution over kernels defined by the data sets in the cluster. We obtain these samples via the Metropolis-Hastings algorithm described in section 10.3.2.

$$p(\boldsymbol{K}_c, \boldsymbol{\theta}_c, \boldsymbol{\sigma}_c^2 | \mathcal{D}, \boldsymbol{Z}) \propto \underbrace{p(\boldsymbol{\theta}_c, \boldsymbol{\sigma}_c^2 | \boldsymbol{K}_c) G_{\text{parent}}(\boldsymbol{K}_c)}_{\text{Prior on kernel}} \times \underbrace{\prod_{c: Z_n = c} p(\mathbf{y}_n | \mathbf{X}_n, \boldsymbol{K}_c, \boldsymbol{\theta}_c, \boldsymbol{\sigma}_c^2)}_{\text{Likelihood of datasets}} \tag{14}$$

Since we are assigning a kernel at each cluster, it is possible for two different clusters to have the same composition but different hyperparameters.

### 10.3.2 Metropolis-Hastings Sampler for Kernel Selection

In the following, we detail the proposal distribution over kernel compositions used by the Metropolis-Hastings sampling algorithm in the paper.

### 10.3.3 Kernel Composition Proposal

Let $K$ by the current kernel composition, which is a sum of kernel components: $K = \sum_{n=0}^{N_K} C_n$, where $C_n \in \mathcal{K}$ is a kernel component and part of the total candidate kernel pool $\mathcal{K}$. Let $N$ be the total number of candidate kernels in the kernel pool. We propose new compositions by randomly adding a component, removing a component, or doing nothing to the current composition. Doing nothing is an option because we may alter the kernel by sampling the hyperparameters, even though the composition remains the same. The $p_{\text{add}}$, $p_{\text{remove}}$, and $p_{\text{nothing}}$ are parameters for the proposal distribution and must sum to 1.

|   |   | $P(\text{add}|K)$ | $P(\text{remove}|K)$ | $P(\text{nothing}|K)$ |
|---|---|---|---|---|
| s | $1 \leq N_K < N$ | 0.2 | 0.4 | 0.4 |
|   | $N_K < 1$ | 0.5 | 0 | 0.5 |
|   | $N_K = N$ | 0 | 0.5 | 0.5 |

If the action is to add or remove a component, we need to choose *which* kernel component, $C'$, to add or remove to the composition. If the action is to add, $C'$ is uniformly sampled from the kernel pool excluding the components that are already in $K$ (there are $N - N_K$ options). If the action is to remove, $C'$ is uniformly sampled from the set of kernel components that currently compose $K$ (there are $N_K$ options).

$$p(C' | \text{action}, K) = \begin{cases} \frac{1}{N - N_K}, & \text{if action } = \text{add} \\ \frac{1}{N_K}, & \text{if action } = \text{remove} \\ 1, & \text{otherwise} \end{cases} \tag{15}$$

The probability of proposing kernel $K'$ from $K$ is then $p(C', \text{action}|K) = p(C'|\text{action}, K)p(\text{action}|K)$. Note that this proposal distribution is not symmetric, and the corresponding acceptance ratio must incorporate $p(K|K')$ and $p(K'|K)$ appropriately.

### 10.3.4 Kernel Hyperparameter Proposal

The proposal distribution for the kernel hyperparameters is a random normal distribution centered at the current *log-transformed* hyperparameter values. We apply the log transformation because the lengthscale, period, amplitude, and observation noise parameters must be positive.

The kernel composition and hyperparameters are closely related; a Linear kernel will not use a hyperparameter on the period, just as a Periodic kernel will not use a "shift" hyperparameter. In order to track the current hyperparameters under a changing kernel composition, the proposal distrbution samples the two independently: $p(K, \theta) = p(K)p(\theta)$. The current hyperparameters are represented by $\mathbf{H} \in \mathbb{R}^{(D+1) \times 4}$, where $D$ is the dimensionality of the data and each column corresponds to a type of hyperparameter – a lengthscale, period, amplitude, shift, and observation noise parameter – respectively. The last row in the matrix is always reserved for the observation noise, which applies to the entire composition (not specific to a dimension). The benefit of this representation is that hyperparameters can be sampled independently of the composition; when the composition is sampled, it imposes a "mask" over the hyperparameter matrix. For example, for a $D = 3$ dimensional data set, a $SE0 + PER1 + LIN2$ kernel composition would impose the following mask:

$$\mathbf{M} = \begin{matrix} 1 & 0 & 1 & 0 & 0 \\ 1 & 1 & 1 & 0 & 0 \\ 0 & 0 & 1 & 1 & 0 \\ 0 & 0 & 0 & 0 & 0. \end{matrix}$$

To reemphasize, each column corresponds to a lengthscale, period, amplitude, shift, and observation noise parameter, respectively. Each row represents a different base kernel (kernel function type and the dimension that it applies to). The final kernel hyperparameters would be represented by applying mask $\mathbf{M}$ to hyperparameter matrix $\mathbf{H}$. In this example, where we have an SE kernel on the 0 dimension of the data, the mask would ensure that the columns containing the relevant hyperparameters, which is the lengthscale (first column) and amplitude (second column), are used with the composition. We recognize that a limitation of this representation is that a composition that requires multiple copies of the same hyperparameter on the same dimension, such as a $PER0 \times PER0$ kernel, could only be represented with one value for the period. However, this is a compact representation which avoids the need to maintain a hyperparameter matrix for each potential kernel composition in the kernel space. To be clear, our proposal method **does not** imply that different types of kernels would use the same lengthscale, since each row is a different base kernel (type of kernel and dimension).

