# OpenReview forum: "Online model selection by learning how compositional kernels evolve"
_TMLR — Accepted by TMLR_

### Review · Reviewer_jRnG · 2023-06-25

**Summary Of Contributions:**

The paper addresses a particular online learning setup, where new models need to be developed quickly for newly appearing tasks (e.g. users). It proposes a kernel learning method (learning kernel compositions) for GPs, which shall be sparse (selects limited number of input features), stable (the features selected do not change much when additional online data arrive), adaptive in complexity (limited complexity at the start when only few data available and gradually increasing), and scalable (not getting unwieldy with increasing time). The method is based on learning a set of ``kernel evolutions'' over readily available pilot data that can then be used in the actual online regime. These evolutions are learned as a Markovian process across the time-steps (online batches of data) with Chinese Restaurant Process across the tasks in the pilot for each evolution step. The properties of the method are clarified on a toy synthetic dataset, and documented on 4 UCI and 2 real medical datasets. The proposed methods performs well in terms of test log-likelihoods, tends to avoid overfitting in small data regimes in the beginning of the data stream, prefers sparse solutions (BIC criteria), and is more stable in variable selection than the presented baselines.


**Audience:**

Yes

**Broader Impact Concerns:**

I do not see any ethical concerns requiring Broader Impact Statement.

**Claims And Evidence:**

Yes

**Requested Changes:**

I recommend addressing the points raised under `Strength and Weakness`.

Other minor comments, that shall, however be addressed before publication:
1. In equation (2) do the weights need to be non-negative?
2. Please clarify briefly the difference between the ARD and SE kernels.
3. Figure 4 - caption mentions lef and middle diagram. There is no middle diagram.
4. You use ARD as a kernel as well as a method. I find this very confusing. Please make it clear in the text. (Also the difference between SE and ARD kernels.)
5. Please Clarify the split to pilot and deployment - nums of tasks in each - in your experiments.
6. Please clarify the total number of time steps in your experiments (length of kernel evolutions)
7. In your online setup there is no separate test set, correct? The only test is the actual online deployment data on which you fit the model parameters and directly evaluate the likelihood (e.g. Figure 7 ``Test'' log likelihood). Please make this clearer to the reader.
8. Page 9 Feasibility paragraph - you refer to figure 6e, I think this is not correct. Please check/correct.
9. Table 1, UCI:Concrete - formatting of +-
10. General formatting of latex quotes - somehow the end quote is difficult to read. Possible to ``correct''?

**Strengths And Weaknesses:**

The explored setup of online multi-task learning (with new tasks appearing) is common in many modern applications and thus is highly relevant. The paper explains clearly the proposed approach, striking a good balance between the necessity to review basic concepts and the expected prior knowledge on the reader's side, making it fairly easy to follow. The experiments as well as the selected tables and graphics clarify and document the performance well.

I am concerned about the scalability - in terms of input features, time points and number of tasks. You mention the challenge of number of tasks in pilot and interaction terms briefly in the the Discussion section, but I believe the scalability considerations deserve more attention. The search for kernel evolutions becomes more complex with higher number of features (and even more with higher number of considered interactions) - how much more complex (some quantification to help user to estimate the usability of the algo)? This also increases the complexity of the search during deployment (BIC over all possible evolutions) - how much?

All experiments are based on a rather limited number of examples. Do you have any recommendation for longer streams? Shall the evolutions be continued or frozen at some point (e.g. when $\mathcal{D_{u,t}}$ is large enough based on some learning theory principals)?

In my understanding, the underlying assumption is some level of distributional stability between the tasks in the pilot and the deployment online setup. I didn't find this in the text. (Or have I missed it? I only found the stability within the stream of a single task.)

There are pieces of information missing (some I found in the appendix) which shall in my view be covered in the main text. See next section for details.

---

> ### Author Response · Authors · 2023-07-18
> **Author response + clarifications**
>
> ***"I am concerned about the scalability - in terms of input features, time points and number of tasks."***
> In our intended applications, we typically have a generous pre-deployment period for model development, but during deployment, we require model adjustments to be data efficient as well as reasonable computationally efficient. Thus, our main goal is to front-load the computation, that is, to learn a model for kernel evolution during pilot training, so that we may more quickly select kernels during deployment. We adjusted the writing in Section 5.1 to reflect the fact that health applications admit pilot training phases that are long enough-- on the order of months to years-- to accommodate training our model at scale. We have clarified in the introduction that our claims of scalability are restricted to efficiency during *deployment only* when compared to existing *kernel composition selection methods*.
>
> We've added exposition and proposed an experiment to address how our method scales with respect to features, time steps and tasks. In general, an increase in features and interactions will increase the total number of base kernels considered. More base kernels means an increase in the number of iterations needed for the Gibbs sampling to converge. During each iteration, the dominant computation of KEM is in repeatedly calculating the model likelihood during training, which involves calculating a GP likelihood for each user (with $N_m$ total users), time step (with $N_t$ total time steps per user). This is $\mathcal{O}((N_m * N_t)^3)$. This computational cost is paid during pre-deployment time. Since we assume a length pre-deployment period during which to learn from pilot data, this order of computation is not a bottleneck for our intended applications. We note that using recent GP approximation methods, such as sparse GP approximations, the cubic term may be reduced to $\mathcal{O}((N_m * N_t)^2 M)$ for some $M << (N_m * N_t)$ [1].
>
> ***Experiments are based on a rather limited number of examples.""***
> In our health settings of interest, such as mobile health, the number of users and timesteps in the pilot study is limited, because of the cost of running such studies. For example, the HeartSteps V1 pilot had $36$ users over $42$ days (5 time steps per day) [2]. During deployment, the goal is rapid personalization with good bias-variance trade-off. That is, we are targeting the window of time where we have few observations from each user. In our experiments, we specifically selected our data sets to be of the same scale we would expect from an mHealth study.
>
> ***Do you have any recommendation for longer streams / increased deployment search.""***
> In high-noise regimes such as health, we expect that the number of kernel compositions identified on a limited amount of pilot data will be small. This is because, in mHealth, each user's trajectory is limited; and, while the CRP theoretically can learn arbitrary numbers of distinct kernel compositions, in practice, the majority of users will fall into a finite number of clusters that our method can characterize well [3].
> Simultaneously, KEM is designed to identify an expressive set of kernel compositions while limiting the number of extraneous compositions, as the number of timesteps increases (i.e. the streams are longer). As such, we do not expect the number of kernel compositions to be a significant computational bottleneck.
>
> We would appreciate it if the reviewer looked at the experiment outlined in our general response, and commented on whether this experiment would help their recommendation or suggest changes to the experiment.

---

> > ### Author Response · Authors · 2023-07-18
> > **Continued Author Response**
> >
> > * *"In equation (2) do the weights need to be non-negative?"* Yes, the weights are non-negative since they represent the kernel amplitude hyperparameter (sometimes called the variance hyperparameter).  However, note that the kernel is still capable of representing negative correlations in the data; for example, a linear kernel with a positive amplitude hyperparameter can represent lines with negative or positive slope. We have updated the description surrounding Eq 2. to reflect this.
> > * *"Please clarify briefly the difference between the ARD and SE kernels."* The ARD is a composition. Precisely, it is a product of SE kernels: $\prod\limits_{d = 1}^D \text{SE}_d$,  where each SE kernel models a dimension $d$ of a data set with $D$ total dimensions.
> > * *"You use ARD as a kernel as well as a method. I find this very confusing. Please make it clear in the text. (Also the difference between SE and ARD kernels.)."* Thank you for pointing out this confusion; we have clarified the ARD baseline in section 6.1 of our revision. The ARD is a kernel composition and a baseline. What we call ``ARD" throughout the experiments is a practical baseline that represents using a GP with a fixed kernel composition-- the ARD kernel-- throughout deployment (no kernel re-selection occurs).
> > * *"Please Clarify the split to pilot and deployment - nums of tasks in each - in your experiments."* In the UCI and health data sets, we split the number of tasks in pilot vs. deployment equally (a 50 / 50 split). In the synthetic dataset, we have 10 pilot tasks and 50 deployment tasks to demonstrate the scalability of our approach during deployment. We have moved the description of the data sets from the Appendix into the main body of the paper in section 6.3.
> > * *"Please clarify the total number of time steps in your experiments (length of kernel evolutions)."* The total number of time steps in our experiments varies depending on the data set. For example, the synthetic data set has $6$ time steps per user while the HeartSteps data set has up to $11$. The number of time steps is included in the data descriptors in Table 1 of the revision.
> > * *"In your online setup there is no separate test set, correct? The only test is the actual online deployment data on which you fit the model parameters and directly evaluate the likelihood (e.g. Figure 7 ``Test'' log likelihood). Please make this clearer to the reader."* Our experiments always report test likelihoods on a test set. Each deployment user $u^*$ has a train and a test set. In all experiments, the train set is the cumulative data $\mathcal{D}_{u^*, t}$ observed about the user up to time $t$. In the synthetic experiments, the test set is composed of $200$ uniformly spaced points along the x-axis from $[0, 20]$.
> >
> >     In the real data experiments, the test set is the data that comes in at time $t+1$. This procedure mirrors the way that models are used during real deployment; a kernel is selected with the cumulative data observed about a user thus far and then used to make predictions for the incoming data, until the next time step when the kernel must be selected again, with the new data incorporated.
> >
> >     We have added explanations about the train and test sets for a deployment user in sections 7.1 and 7.2.
> > * All typos and formatting comments have been amended in the revision.
> >
> > References
> > * [1] Titsias, M. (2009, April). Variational learning of inducing variables in sparse Gaussian processes. In Artificial intelligence and statistics (pp. 567-574). PMLR.
> > * [2] Klasnja, P., Smith, S., Seewald, N. J., Lee, A., Hall, K., Luers, B., ... & Murphy, S. A. (2019). Efficacy of contextually tailored suggestions for physical activity: a micro-randomized optimization trial of HeartSteps. Annals of Behavioral Medicine, 53(6), 573-582.
> > * [3] Miller, J. W., & Harrison, M. T. (2013). A simple example of Dirichlet process mixture inconsistency for the number of components. Advances in neural information processing systems, 26.
> > * [4] Ghosh, S., Kim, R., Chhabria, P., Dwivedi, R., Klasjna, P., Liao, P., ... & Murphy, S. (2023). Did we personalize? Assessing personalization by an online reinforcement learning algorithm using resampling. arXiv preprint arXiv:2304.05365.

---

> > > ### Comment · Reviewer_jRnG · 2023-07-20
> > >
> > > Thanks, seems fine now.

---

> > ### Comment · Reviewer_jRnG · 2023-07-20
> >
> > Hi, thanks.
> >
> > 1. Scalability - yes, the experiment would help
> >
> > 2. Low number of examples - ok, I understand now that the experimental setup corresponds to the low-data regime expected in practice. This should be made clearer in the text (if not already done in the updated version). Also some considerations about when is this method actually no longer useful (higher data regimes?) should be mentioned.
> >
> > 3. You haven't answered my concern about the stability assumption (I think you by mistake mixed the replies to the reviewers :) )

---

> > > ### Author Response · Authors · 2023-07-20
> > > **Response to stability assumption**
> > >
> > > Apologies about missing point 3; we wrote a response / updated the manuscript but didn't post it here!
> > >
> > > It is true that we assume distributional stability from pilot to test users; we made this assumption more explicit, with justification, in section 4 of our revision.  This is a reasonable assumption in health, because the pilot data (and trial) are intentionally designed to be drawn from the same target population as the deployment population. For example, an earlier and follow-up version of the HeartSteps trial both targeted adults newly diagnosed with stage 1 hypertension in the Seattle area [4].
> > >
> > > If the population were to shift from pilot to test, it is possible to update KEM with the incoming test data. The Chinese Restaurant Process allows enough flexibility so that we may discover new kernel compositions for the new test data. However, retraining would incur the scalability issues mentioned above, which are typically avoided in deployment because we fix the model.

---

### Review · Reviewer_t2mc · 2023-07-08

**Summary Of Contributions:**

In a multi-task setting, authors propose a kernel method to learn an appropriate kernel evolution that would select more complex kernels when more training data is available for a task.


**Audience:**

Yes

**Claims And Evidence:**

Yes

**Requested Changes:**

-the number of observations seen for users in the Piolot training phase should be similar to the ones expected in the online phase, otherwise the kernels may result too simple. This is another strong assumption that may be discussed in more detail.

-related works: When you discuss e.g. Zhang et al., 2019 you mention that those methods are suited for non-stationary functions. Assuming a function as stationary in online learning is a strong assumption. Can you motivate more in-depth this choice? Also, is it possible to apply the method proposed by Zhang et al., 2019 also in a stationary setting? If so, an experimental comparison would be nice.

-The proposed method does not perform well on the UCI: Fires dataset. Can authors analyse this dataset more in-depth and try to understand why this happens? e.g. which are the dataset characteristics that make the proposed method not perform as well as the baselines?

-Comparison with other models: neural networks are commonly adopted for sequential data, e.g. LSTMs. With NNs it is easy to perform multi-task, transfer or meta learning (MAML as mentoned by authors) and, with the proper hardware, they tend to be quite fast. Can you include this comparison or motivate why it is not reasonable to include it? Authors explain it with a lack of sparsity, that in my understanding would mean a lack of explainability, which brings me to the last point in the next comment. However, it would be nice to see how these methods perform, also to understand the difference in predictive performances we are paying for a sparse solution.

-you mention sparsity as a requirement for high-stakes applications. I guess this is directly related to the interpretability of the resulting models. Can you discuss more in depth this characteristic of the proposed method? Can it identify the features that would be relevant for experts, so they can trust the system?

**Strengths And Weaknesses:**

+Given the strict setting discussed by the authors, the method seems reasonable and shows good experimental results.

-The limitations of the proposed method should be discussed in more detail, see next section.

---

> ### Author Response · Authors · 2023-07-18
> **Author response + clarifications**
>
> * *"The number of observations seen for users in the pilot training phase should be similar to the ones expected in the online phase, otherwise the kernels may result too simple. This is another strong assumption that may be discussed in more detail."* Thank you for pointing out this assumption; we have added it in our (new) list of assumptions in section 4 of our revision.
> * *"Related works: When you discuss e.g. Zhang et al., 2019 you mention that those methods are suited for non-stationary functions. Assuming a function as stationary in online learning is a strong assumption. Can you motivate more in-depth this choice?"* Stationarity is a strong assumption that is motivated by our health setting. Though realistically, health settings are non-stationary, stationarity is often assumed because the noisiness and sparsity of the data make it difficult to model this non-stationarity directly online. For example, many online Reinforcement Learning algorithms in health must make use of a stationarity assumption to preserve the Markovian property of Markov Decision processes [1, 2].  We have brought more attention to this strong assumption in Section 4 of our revision.
> * *"Also, is it possible to apply the method proposed by Zhang et al., 2019 also in a stationary setting? If so, an experimental comparison would be nice."* The method proposed by Zhang et al. does not include a search over kernel compositions, and therefore, would not be a suitable comparison in our setting.
>     Specifically, the generative process proposed by Zhang et al. (with notation updated to reflect our own) is as follows:
>     \begin{align}
>         \alpha \sim \text{Gamma}(a_0, b_0); \\
>
>         z_t \sim CRP(\alpha); \\
>
>         \theta_c \sim p(\theta_c); \\
>
>         \sigma^2_c \sim p(\sigma^2_c); \\
>
>         f_{t} \sim \text{GP}(0, K_{\theta_c}); \\
>
>         x_t \sim p(x_t | z_t) \\
>
>         y_{t} \sim \mathcal{N}(f_{t}(x_t),\sigma^2_c) ,
>     \end{align}
>     where $(X_c, y_c) = (x_t, y_t: z_t = c)$ represent the data associated with the mixture $c$.
>
>     Like our KEM, this model includes an infinite mixture over GPs by using a Chinese Restaurant Process. Unlike ours, the infinite mixture is over GPs *with the same kernel (a SE kernel) but different hyperparameters, $\theta_c$ and $\sigma_c^2$*. Their mixture *is not* over different kernel compositions; there is no distribution over $K_c$. As a result, the Zhang 2019 method does not help us decide how to (sparsely) select kernel compositions for users with online data.
>
>     We have moved the reference to a more appropriate section (a paragraph down) in the related works and expanded on its description.
> * *"The proposed method does not perform well on the UCI: Fires dataset. Which are the dataset characteristics that make the proposed method not perform as well as the baselines?"*     Fire is the simplest data set, so all methods perform “equally well.” We have added a summary of the datasets in Table 1, followed by a discussion of their characteristics. Furthermore, we have elaborated on the performance of all methods on Fires in the ``Adaptive complexity'' paragraph of section 7.2.

---

> ### Author Response · Authors · 2023-07-18
> **Author response cont.**
>
> * *"Comparison with other models: sequential NNs, MAML."* Our paper scope concerns Gaussian Processes because of their ability to represent uncertainties and provide an interpretable way to map to domain knowledge (through inspecting the learned kernels). Both are important qualities for models of healthcare data. Within GPs, we focus on the problem of efficiently selecting high-performing kernel compositions. As the reviewer notes, we focus on compositions because interpretability is desired.
>
>     We have revised the paper to more clearly emphasize the scope of methods under consideration: *compositional kernel selection* for *GP regression*. Furthermore, we have clarified that our claims regarding scalability apply to *other CKS methods* for GPs during deployment.
>
>     We hypothesize two cases where a MAML-type approach using neural networks would not work well in our setting. First, the few-shot transfer may not behave well for noisy, low data settings (specifically, with respect to our MIMIC and HeartSteps data sets). Second, supposing this transfer performs well, it would still sacrifice the interpretability that comes with uncertainties and sparse compositions. We could implement a NN-MAML baseline to explore these hypotheses. However, we hope that reducing the claims to concern compositional kernel selection for GPs will remove the need to test against NNs and are open to further discussion.
> * *"Can you discuss more in depth this [interpretability] characteristic of the proposed method? Can it identify the features that would be relevant for experts, so they can trust the system?"* Selecting *sparse kernel compositions* (as opposed to using ARD kernel, one of our baselines) allows us to identify relevant features *and* their relationships to the target data. For example, in 8b we identified that peak inspiratory pressure has a non-linear relationship with platau pressure, while respiration rate has a linear one. Such relationships are important to identify so that they can be inspected by an expert. Furthermore, there is work that suggest human's intuitive understanding of complex functions is compositional [3]; which supports the goal of identifying compositions in our data.
>
>     Our method specifically selects kernel *evolutions*. This gives us additional information about what compositions are important and shared for all users (e.g. compositions that appear early in the trajectories of the selection model, such as the linear kernel in 3c) and what compositions are applicable to only a subset of users (e.g. compositions that appear later in the trajectories of the selection model, such as the SE and LIN + PER kernel in 3c).
>
>
> References
> * [1] Trella, A. L., Zhang, K. W., Nahum-Shani, I., Shetty, V., Doshi-Velez, F., & Murphy, S. A. (2022). Designing reinforcement learning algorithms for digital interventions: pre-implementation guidelines. Algorithms, 15(8), 255.
> * [2] Liao, P., Greenewald, K., Klasnja, P., & Murphy, S. (2020). Personalized heartsteps: A reinforcement learning algorithm for optimizing physical activity. Proceedings of the ACM on Interactive, Mobile, Wearable and Ubiquitous Technologies, 4(1), 1-22.
> * [3] Schulz, E., Tenenbaum, J. B., Duvenaud, D., Speekenbrink, M., & Gershman, S. J. (2017). Compositional inductive biases in function learning. Cognitive psychology, 99, 44-79.

---

### Review · Reviewer_U5o6 · 2023-07-08

**Summary Of Contributions:**


The authors address the task of multi-task Gaussian Process regression and present a method that selects compositional kernels in an online setting. The idea is to enable the evolution of the kernel as more users are observed; this allows the method to handle the bias-variance trade-off better. The new method is termed Kernel Evolution Model (KEM) and is motivated by online multitask applications in healthcare. The authors evaluate KEM on synthetic and real data, demonstrating its flexibility and stability.


**Audience:**

Yes

**Broader Impact Concerns:**

No ethical concerns

**Claims And Evidence:**

Yes

**Requested Changes:**

I think the paper could be organized better, specifically moving the related work forward and briefly introducing the problem in the introduction to help the reader connect to the problem.
The example in Figure 1 is unclear; it is hard to identify differences between the blue and red results. Can you quantify this numerically? Also, the explanation of this example could be improved; for example, each element should be explained in the caption.
Commas are missing after some of the equations (eq. 1…)
The authors discuss the bias-variance tradeoff several times; intuitively, I understand why it provides benefits here, but is there a way to demonstrate this?
The method is designed for sparse selection; why not demonstrate the feature selection capabilities of the method as in Figure 8 but with synthetic data where the informative features are known?
Figure 6e is introduced before figures 4 and 5; is e a typo?
ARD performs very similarly to KEM but much faster in the synthetic example. Are there examples in which the merits of the approach are more significant?
ARD uses a constant kernel, so why present it in Figure 5?
What are the properties of the real datasets?
What is the model's accuracy (and baselines ) for the real data?


**Strengths And Weaknesses:**

The authors address an interesting problem in ML, where the complexity of the kernel can evolve as more users are added. Overall, the English level is satisfactory.
While the empirical results demonstrate the merits of the approach, I have some concerns about the paper and think it should be revised significantly prior to publication.
Specifically, the presentation should be dramatically improved. The empirical evaluation is only focused on relatively small datasets and only one extremely small synthetic evaluation.

---

> ### Author Response · Authors · 2023-07-18
> **Author response + clarifications**
>
> * *"The empirical evaluation is only focused on relatively small datasets."* In our health settings of interest, such as mobile health, the number of users and timesteps is limited. Our data sets are of the same scale we would expect from an mHealth pilot / feasibility study. For example, the HeartSteps V1 pilot had $36$ users over $42$ days (5 time steps per day) [1].  Furthermore, the dimensionality of the data is also reflective of the dimensionality of the data used for predictions in these micro-randomized trials [2, 3]. We have made our motivation for selecting the data sets of our current scale in section 6.3.
> * *"I think the paper could be organized better."* We have made a number of changes to the organization and exposition of the paper to better highlight our problem setting, assumptions and justifications. Please see the revised version for these changes.
> * *"The example in Figure 1 is unclear. Also, the explanation of this example could be improved; for example, each element should be explained in the caption"* In our revision, we added the test likelihoods to the sub-figures in Figure 1. The test likelihood quantifies the gaussian process's ability to predict the true function, with the given kernel; higher is better. In this demonstrative example, our method (the right two plots) have higher test likelihood than the baseline method (the left two plots). We expanded on the caption of Figure 1 in our uploaded revision.
> * *"The bias-variance tradeoff... intuitively, I understand why it provides benefits here, but is there a way to demonstrate this?"*  Our experiments demonstrate the bias-variance tradeoff in Figures 4a (synthetic data) and 7 (real data). These results show (as we expect), that our selection method has good predictive performance when there is little data (due to us incurring bias in order to lower variance), as well as when there is sufficient data (due to us decreasing bias). We have updated our writing in section 7.1 to explicitly tie the good predictive performance in Figure 4 to KEM's ability to manage the bias-variance trade-off.
> * *"The method is designed for sparse selection; why not demonstrate the feature selection... why not demonstrate the feature selection with synthetic data where the informative features are known?"* Thank you for the suggestion; we believe such an experiment would more clearly demonstrate our method's ability to select the correct *features.* We would appreciate it if the reviewer would look at the experiment outlined in our general response, which considers synthetic data with multiple dimensions, and comment on whether this experiment would help their recommendation or suggest changes to the experiment.

---

> ### Author Response · Authors · 2023-07-18
> **Author response cont.**
>
> * *"ARD performs very similarly to KEM."* In Figure 7, there are several data sets for which ARD performs the worst (UCI: Energy, UCI: Housing, MIMIC). We show that this is because ARD is overfitting the lengthscale hyperparameters (one per dimension). In general, we believe ARD kernel does well on synthetic data because it corresponds to selecting a SE0 kernel at every time step (ARD is a kernel composition, where an SE kernel has been applied to each data dimension; in the synthetic case, there is only one dimension). This is a relatively simple composition for which the overfitting is minimized. However, ARD's overfitting becomes more apparent on the multi-dimensional data. We have highlighted this difference in ARD performance in 1-dimension vs. multiple dimensions in the results section.
> * *"Why present it [ARD] in figure 5?"* We included the ARD kernel for completeness, to avoid confusing the reader by excluding a baseline. We are happy to remove ARD from the plot if this is in fact more confusing.
> * *"What are the properties of the real datasets?"* We have moved the summary of data sets from the appendix to Section 6.3. Additionally, we have added a description of the datasets with respect to two characteristics: level of user heterogeneity and noise. We are particularly interested in the performance of the kernel selection methods on the high noise, high heterogeneity data, since these characteristics describe the health setting.
> * *"What is the model's accuracy (and baselines) for the real data?"* In Figure 4a and 7, we report the test likelihoods, which quantify our Bayesian model's accuracy with respect to its mean and uncertainty predictions. All baselines described in Section 6.1 are used for the real data, withholding MH due to the infeasibility of scaling in deployment.
>
> References
> * [1] Klasnja, P., Smith, S., Seewald, N. J., Lee, A., Hall, K., Luers, B., ... & Murphy, S. A. (2019). Efficacy of contextually tailored suggestions for physical activity: a micro-randomized optimization trial of HeartSteps. Annals of Behavioral Medicine, 53(6), 573-582.
> * [2] Trella, A. L., Zhang, K. W., Nahum-Shani, I., Shetty, V., Doshi-Velez, F., & Murphy, S. A. (2022). Designing reinforcement learning algorithms for digital interventions: pre-implementation guidelines. Algorithms, 15(8), 255.
> * [3] Liao, P., Greenewald, K., Klasnja, P., & Murphy, S. (2020). Personalized heartsteps: A reinforcement learning algorithm for optimizing physical activity. Proceedings of the ACM on Interactive, Mobile, Wearable and Ubiquitous Technologies, 4(1), 1-22.

---

> > ### Comment · Reviewer_U5o6 · 2023-08-03
> > **Response to authors**
> >
> > I want to thank the authors for considering the reviewers' suggestions and for improving their paper.
> > They address most of my concerns. Regarding:
> > "We would appreciate it if the reviewer would look at the experiment outlined in our general response, which considers synthetic data with multiple dimensions, and comment on whether this experiment would help their recommendation or suggest changes to the experiment"
> > Indeed, I support adding these experiments; they will improve the paper and help readers understand and use the method.

---

### Decision · Action_Editors · 2023-08-12

**Recommendation:** Accept with minor revision

**Comment:**

Dear authors,

All three reviewers have praised the interest and soundness of your contributions. I am happy to recommend acceptance provided that you can satisfactorily expand some of the comments on the limitations (see paragraph below), and include the new experiments that you suggested to better study how scalable your method is.

One of the reviewer wrote to me (not verbatim, these are my words):
An argument for a minor revision is issues related to scalability (number of features, expected interactions, users, time-steps). The paper describes the setting for when the method works well (the specific mHealth setup with large pilot small deployment, etc.) but does not provide enough guidance to be able to judge when it no longer should be used. The authors suggested an experiment to better explore the scalability of their method and this should included and commented upon before publication.

Congratulations on a fine piece of work!

Benjamin

**Audience:**

I am confident the paper will be of interest to a broad part of TMLR's audience.

**Claims And Evidence:**

Yes, all claims are supported by convincing evidence.

---

> ### Author Response · Authors · 2023-08-21
> **Thank you for the decision**
>
> Dear action editor and reviewers,
>
> Thank you for the thoughtful responses and the decision! We are working on our revision with the additional experiment.

---

> ### Author Response · Authors · 2023-10-25
> **Uploaded final revision with scalability experiments**
>
> We thank the reviewers and action editors for the constructive feedback regarding our paper!
>
> We included an experiment to explore the scalability of our approach during pilot training in section 7.3.
> In summary, we found that runtime in pilot training scales better to more users (scales linearly) than to more timesteps (scales exponentially). This implies that our method is best suited for pilot data that has either a high number of users and limited time steps, or a larger number of batches per time step with fewer total time steps. The number of features (and size of the base kernel set) did not affect the amount of time it took per training iteration in our method; however, we expect that pilot data with a larger number of features will require more iterations overall.
>
> Finally, we addressed each reviewer's suggestions during the discussion period and amended the final manuscript as noted in each of the individual responses.